evolution/genetics/genomics

adaptation, ecotype, convergent evolution, parallel evolution, polyploidy, introgression

**Author for correspondence:**
Tyler K. Chafin
e-mail: tylerkchafin@gmail.com

# Parallel introgression, not recurrent emergence, explains apparent elevational ecotypes of polyploid Himalayan snowtrout

Tyler K. Chafin[1,2], Binod Regmi[1,3], Marlis R. Douglas[1], David R. Edds[4], Karma Wangchuk[1,5], Sonam Dorji[5], Pema Norbu[5], Sangay Norbu[5], Changlu Changlu[5], Gopal Prasad Khanal[5], Singye Tshering[5] and Michael E. Douglas[1]

[1]Department of Biological Sciences, University of Arkansas, Fayetteville, AR 72701, USA
[2]Department of Ecology and Evolutionary Biology, University of Colorado, Boulder 80309, USA
[3]National Institute of Arthritis, Musculoskeletal and Skin Diseases (NIAMS), National Institutes of Health, Bethesda, MD 20892, USA
[4]Department of Biological Sciences, Emporia State University, Emporia, KS 66801, USA
[5]National Research and Development Centre for Riverine and Lake Fisheries, Ministry of Agriculture and Forests, Royal Government of Bhutan, Haa, Bhutan

TKC, 0000-0001-8687-5905; MED, 0000-0001-9670-7825

The recurrence of similar evolutionary patterns within different habitats often reflects parallel selective pressures acting upon either standing or independently occurring genetic variation to produce a convergence of phenotypes. This interpretation (i.e. parallel divergences within adjacent streams) has been hypothesized for drainage-specific morphological 'ecotypes' observed in polyploid snowtrout (Cyprinidae: *Schizothorax*). However, parallel patterns of differential introgression during secondary contact are a viable alternative hypothesis. Here, we used ddRADseq ($N = 35\,319$ *de novo* and $N = 10\,884$ transcriptome-aligned SNPs), as derived from Nepali/Bhutanese samples ($N = 48$ each), to test these competing hypotheses. We first employed genome-wide allelic depths to derive appropriate ploidy models, then a Bayesian approach to yield genotypes statistically consistent under the inferred expectations. Elevational 'ecotypes' were consistent in geometric morphometric space, but with phylogenetic relationships at the drainage level, sustaining a hypothesis of independent emergence. However, partitioned analyses of

phylogeny and admixture identified subsets of loci under selection that retained genealogical concordance with morphology, suggesting instead that apparent patterns of morphological/phylogenetic discordance are driven by widespread genomic homogenization. Here, admixture occurring in secondary contact effectively 'masks' previous isolation. Our results underscore two salient factors: (i) morphological adaptations are retained despite hybridization and (ii) the degree of admixture varies across tributaries, presumably concomitant with underlying environmental or anthropogenic factors.

# 1. Introduction

Selection for local environmental conditions can drive rapid evolution and occasionally does so within parallel systems to generate 'convergent' adaptations [1]. This, in turn, can promote the emergence of unique ecotypes, or even novel species [2–4]. Adaptations within continuous ecological gradients can also occur, even in the absence of geographic isolation, and are commonly attributed to selection acting upon existing variation (either ancestral/standing [5,6], or that acquired through hybridization [7–11]). However, these represent but two of several scenarios through which such patterns can be generated [12–14].

Divergent selection in ecological gradients can also facilitate speciation, even when gene flow is ongoing [15–17]. A hallmark of this process is the formation of genomic 'islands of divergence', with selection and its cumulative effects serving to counterbalance homogenizing gene flow [18–22]. These so-called 'islands' may then expand via a hitchhiking mechanism, such that divergence is also initiated within linked genomic regions [23–26]. However, heterogeneous genomic divergence can also arise via entirely different processes, to include those wholly unrelated to primary divergence. This, in turn, introduces an analytical dilemma, in that the signatures of one can either obfuscate or instead emulate that of the other [27–29].

The emergence of parallel ecotypes is often manifested phylogenetically as clusters within study sites or regions, rather than among ecophenotypes [30]. However, an alternative is that ecological adaptations instead evolve via isolation followed by subsequent genetic exchange, such that genomic loci are now juxtaposed across both distributions and genomes [12,31,32]. 'Genomic islands' as well as phylogenetic patterns are then generated similar to those manifested by parallel divergence-with-gene-flow (per above). This can occur, for example, when selection constrains introgression within localized genomic regions that underlie adaptation [33]. Genome-wide homogenization is the result, with ancestral branching patterns (e.g. those uniting ecotypes) now restricted to regions where permeability to gene flow is reduced by selection and/or low recombination [34,35]. However, the singular origin of an adaptive allele spread selectively via localized introgression can also yield conflicting genomic patterns [36–38].

Together, this implicates four distinct scenarios (figure 1) that could generate genomic landscapes compatible with those expected under parallel ecotype emergence. Yet only two of these necessitate divergence-with-gene-flow, e.g. with selection acting in a repetitive fashion on either (a) standing genetic variation or (b) independent *de novo* mutations. By contrast, two alternative scenarios involve adaptive variability accumulated while in isolation, followed by secondary gene flow among dispersing ecotypes. This then results in (c) selective filtering against a background of genomic homogenization; or (d) the selective introgression of adaptive alleles into novel populations (figure 1). Thus, while all are effectively operating in 'parallel' (e.g. in separate river drainages or habitat patches), their relationship to the diversification process differs substantially. This presents a challenge for the interpretation of such patterns, in that those superficially similar may in fact reflect markedly different processes that potentially act in concert [39–41]. A highly localized genomic architecture underlying ecotypes further complicates this situation [42].

## 1.1. Can parallel emergence be discriminated from parallel introgression?

We posit these factors can indeed be discriminated by predicting the ancestries for replicated 'pairs' of ecotypes distributed among sites (although patterns may also depend upon migration rates among populations [43]). For hypotheses of 'independent emergence', we would predict that genealogies with regions encapsulating targets of divergent selection (i.e. scenarios (a) or (b); figure 1) would lack shared ancestry among ecotypes, reflecting their independent origins. In a similar vein, a failure to

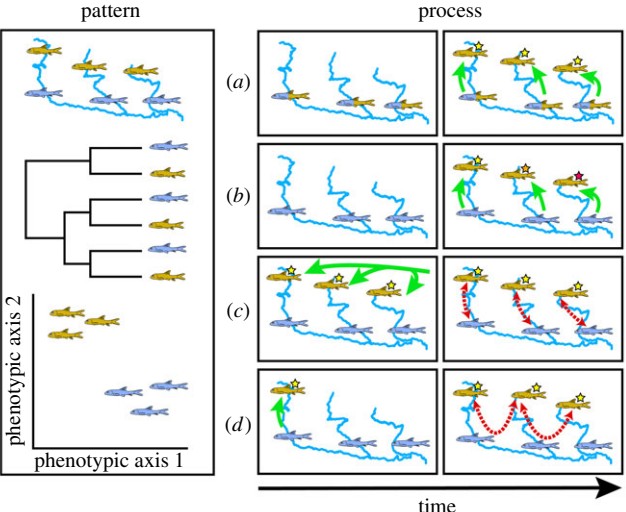

**Figure 1.** Pattern and process in the formation of apparent ecotypes. Ecotypy is often inferred via a juxtaposition of discordant spatial, phylogenetic and phenotypic patterns (left; differentially adapted forms represented in gold and purple). Evolutionary scenarios (right) that can generate these patterns include (*a*) parallel selection upon standing variation; (*b*) recurrent *de novo* adaptation via independent mutation; (*c*) hybridization among co-occurring ecophenotypes following a common origin or (*d*) the singular origin of an adaptive mutation, which is then spread via adaptive introgression. Green arrows indicate dispersal/colonization, red-dashed arrows indicate hybridization/introgression.

correlate with unlinked targets of selection would also be expected [12]. Thus, in the case of repeated selection upon standing genetic variation, adaptive alleles will be identical-by-descent but with flanking regions excluded (given that populations will lack identical, independently fixed haplotypes) [13,14]. Furthermore, in a scenario of independent emergence, mutations underlying adaptation may occur at different loci altogether.

By contrast, a 'divergence first' model with subsequent secondary contact (scenarios (*c*) or (*d*); figure 1) would imply that ancestries of ecotypes are both correlated with and shared among ecotypes at those loci targeted for selection. However, genomic patterns can be difficult to disentangle from those expected under a scenario of adaptive introgression (scenario *d*). This is because gene flow during secondary contact may effectively 'swamp' divergence developed in isolation [12,44]. In the absence of genomic resources (a frequent scenario for non-model organisms), additional clarification can often be inferred from the biogeographic context.

## 1.2. A case study involving *Schizothorax*

We here investigate the source of adaptive genetic variation among differentially adapted pairs of rheophilic freshwater fishes occupying elevational gradients in Himalayan tributaries of the Ganges and Brahmaputra rivers. Our study group (snowtrout; Cyprinidae: *Schizothorax* spp.) is of interest in that it displays a legacy of whole-genome duplication, with variable adaptations to high-elevation habitat (both phenotypic and life-historic) [45–48]. Yet, in previous studies, patterns of convergence are also broadly apparent [49,50]. Additionally, the rapid formation of multiple 'species flocks' has occurred within unique and isolated alpine lake habitats [51–55], with recurrent divergences as an emerging consensus [51,56]. Here, two Himalayan species are of particular interest: *Schizothorax progastus* and *S. richardsonii*. Both are distributed along with an elevational gradient, with generalized morphologies within tributaries of the Ganges and Brahmaputra rivers converging upon 'blunt-nosed' (*S. richardsonii*) in upstream reaches and 'pointed-nosed' (*S. progastus*) in downstream reaches [52,57–59].

The 'pointed-nosed' ecophenotype displays a more terete, streamlined shape, and gradually replaces the 'blunt-nosed' form longitudinally along with an elevational gradient, a pattern seemingly replicated in each of several collateral, southward-flowing tributaries in Nepal, which in turn suggests the presence of ecological non-exchangeability [60]. Recent phylogenetic analyses based on mtDNA failed to support the current taxonomy, with relationships occurring instead at the drainage level [56]. Despite this, morphologies are consistent across drainages [51], suggesting an independent emergence of 'blunt-nosed' phenotypes (i.e. those associated with *S. richardsonii*) within each highland drainage.

Morphologically convergent ecotypes are also replicated within elevational gradients of Bhutan [61], again suggesting the formation of parallel ecotypes. This suggests the potential for rapid evolution, particularly when juxtaposed with the adaptive radiation by *Schizothorax* within isolated Rara Lake of Nepal [51,53].

Ploidy has been hypothesized as facilitating diversification within cyprinid fishes [62], with a rapid, positive shift in net diversification occurring within three predominantly polyploid subfamilies: Torinae, Schizopygopsinae and Schizothoracinae, with the most profound effect in the latter. Of particular interest, all three subfamilies are endemic to the Himalayan and Qinghai-Tibetan Plateau (QTP) regions and possess life-history specializations for high-elevation existence [63]. Thus, the diversification of polyploid cyprinids on the QTP and adjacent regions is seemingly linked to extensive orogeny followed by periods of marked climate change [62,64–66].

However, population genomic methods are currently limited in their capacity to gauge how polyploidy has driven the adaptation by *Schizothorax* to elevational gradients. Studies involving non-model polyploid species (as herein) must employ methodological paradigms that assume diploidy, yet with fundamentally divergent theoretical expectations [67,68]. An additional complication involves the varying degrees of divergence and/or conservation found among ohnologs (i.e. duplicated loci originating from whole-genome duplication) that potentially occur in those species at intermediate stages of re-diploidization [69]. One positive is that models suitable for genotyping polyploid or mixed-ploidy data [70,71] now incorporate short-read sub-genomic methods (e.g. RADseq and related methods).

The identification of genomic adaptations to elevation in *Schizothorax* has two trenchant stumbling blocks: drivers of morphological and phylogenetic discordance are not only numerous, but also constrained by polyploidy. To compensate, we combine novel statistical models and robust genotyping of non-model polyploids with expectations regarding how ancestries should be distributed across the genome. This allows us to address several questions regarding the evolution of diverse *Schizothorax* lineages seemingly replicated in the Himalayan drainages of Nepal and Bhutan:

1. Does morphological and phylogenetic discordance in parallel Himalayan elevational gradients stem from parallel 'independent emergence (figure 1*a*,*b*), or secondary/ongoing gene flow (figure 1*c*,*d*)?
2. If (*a*) or (*b*), do ecotypic pairs in replicated drainages show evidence of co-divergence, thus indicating a shared underlying biogeographic process (e.g. periods of QTP uplift)?
3. If (*c*) or (*d*), is there evidence for selection-mediated differential introgression?
4. Finally, is there evidence for either rapid co-divergence or hybridization as playing a role in the formation of lacustrine ecotypes?

# 2. Methods

## 2.1. Sampling and DNA preparation

Tissue samples ($N = 96$) represent six *Schizothorax* species distributed throughout tributaries of the Brahmaputra and Ganges rivers (Bhutan and Nepal, respectively) (figure 2*a*–*c*). Of these, $N = 48$ Nepali specimens were obtained from the University of Kansas Natural History Museum (KUNHM) (figure 2*b*; electronic supplementary material, table S1 and S2) and represent two riverine species ('high-elevation' *S. richardsonii* ($N = 16$) and 'low-elevation' *S. progastus* ($N = 8$)). These were sampled from each of the three major Nepali drainages (Kali Gandaki, Koshi and Karnali), except for *S. progastus* from Karnali. The remaining Nepali samples ($N = 24$) represent a land-locked lacustrine radiation endemic to Lake Rara [53]. These are *S. macrophthalmus* ($N = 8$), *S. raraensis* ($N = 8$) and *S. nepalensis* ($N = 8$).

Bhutanese *Schizothorax* ($N = 48$) were sampled from three major Bhutanese drainages (Wang Chhu, Punatsang Chhu and Mangde Chhu: figure 2*c*; electronic supplementary material, table S1) and identified as *S. progastus* and *S. richardsonii* based upon morphological diagnoses of vouchered specimens [52]. However, subsequent mtDNA sequence analysis has putatively identified Bhutanese *S. richardsonii* as another convergent high-elevation specialist, *S. oconnori* [56], a species typically found on the QTP [72,73] within high-elevation tributaries of the upper Brahmaputra River (herein referred to as the Yarlung-Tsangpo River). They are subsequently referred to herein as *S. cf. oconnori* (figure 2*c*).

Tissues were processed using the Qiagen DNeasy Blood and Tissue Kit (Qiagen, Inc.), following manufacturer's protocols. Extracts were evaluated for the presence of high-molecular weight DNA using gel electrophoresis (2% agarose) and quantified at 2 µl per sample in 200 µl assays using Qubit

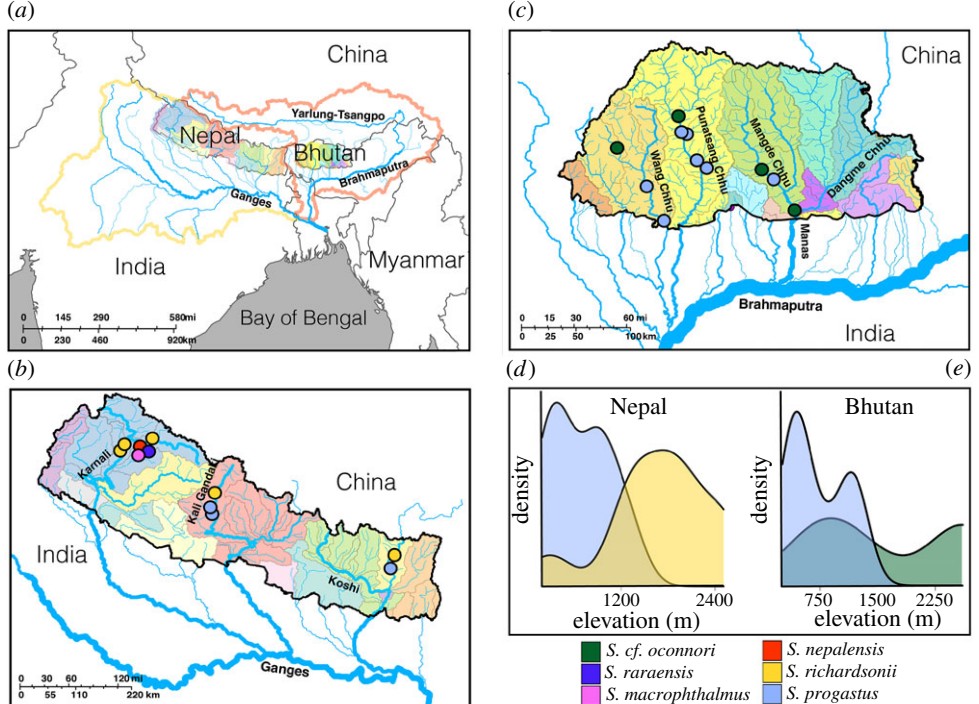

**Figure 2.** Sampling locality information for $N = 96$ Schizothorax spp. selected for ddRAD sequencing. Sites represent major Himalayan tributaries of the Brahmaputra and Ganges rivers from Nepal and Bhutan (*a*). Samples for five species (*S. macrophthalmus*, *S. nepalensis*, *S. progastus*, *S. raraensis* and *S. richardsonii*) were collected from 11 sites in the major drainages of Nepal (Karnali, Gandaki and Koshi) (*b*). Major drainages of Bhutan (Wang Chhu, Punatsang Chhu and Mangde Chhu) were represented by two species (*S. progastus* and *S. cf. oconnori*) collected from 11 localities (*c*). Schizothorax progastus from both Nepal (*d*) and Bhutan (*e*) were collected from relatively lower elevations (approx. 600–1200 m), whereas *S. richardsonii* (Nepal; *d*) and *S. cf. oconnori* (Bhutan; *e*) were generally found greater than 1200 m.

broad-range DNA fluorometry assays (Thermo Fisher Scientific). DNA (500–1000 ng of DNA per sample in 50 µl reactions) was then fragmented using a restriction double-digest [74] with *Pst*I (5′-CTGCAG-3′) and *Msp*I (5′-CCGG-3′). Digests were subsequently visualized on 2% agarose gels, purified using AMPure XP beads (Beckman Coulter, Inc.) and again quantified via Qubit fluorometer.

Samples were then standardized at 100 ng of digested DNA and ligated in 30 µl reactions using T4 DNA ligase (New England Biolabs, Inc.) following manufacturer's protocols. Barcoded oligonucleotide adapters were designed and employed following Peterson *et al.* [74]. After a second AMPure XP purification, samples were multiplexed in groups of $N = 48$ and size-selected at 350–400 bp (excluding adapters), using a Pippin Prep automated system (Sage Sciences). Adapters for Illumina sequencing were subsequently extended via a 10-cycle PCR with Phusion high-fidelity DNA polymerase (New England Biolabs, Inc.). Final reactions were purified via AMPure XP beads and standardized per submission requirements of the DNA Core Facility (University of Oregon Genomics & Cell Characterization Facility, Eugene, OR USA). Additional quality control at the core facility included fragment size analysis (to confirm successful amplification and the absence of artefacts) and qPCR (to assess the proportion of sequenceable library). Sequencing pooled two $N = 48$ multiplexed libraries into a single lane of $1 \times 100$ sequencing on the Illumina HiSeq 4000.

## 2.2. Data filtering and ploidy-aware assembly

Raw reads were demultiplexed and filtered for initial assembly using IPYRAD [75]. Reads having more than zero barcode mismatches or more than five low-quality bases were discarded and adapter sequences trimmed using the 'strict' option. Two assemblies were performed: one *de novo* (at an 85% identity threshold) and one using bwa to align against assembled transcriptome data (hereafter 'transcriptome-guided') [76]. Because the genotyping model in IPYRAD assumes diploidy [77], we used relaxed settings so as to retain assembled paralogs (e.g. electing for more stringent filtering following ploidy-aware genotyping; see below). These included a minimum depth threshold of only six, allowing 20 heterozygous sites per consensus locus, with up to four alleles at a given locus within an individual.

To explore potential ploidy variation in our samples, we first employed BWA [78] to realign raw reads against the candidate locus catalogue generated in our *de novo* assembly. For ploidy model selection, we then computed allelic read depths for bi-allelic sites using a de-noising procedure (NQUIRE; [79] with expectations that ploidies represent allelic depths as follows: at approximately 50% in a diploid bi-allelic heterozygote (i.e. an approximate $50:50$ representation of each allele in an AB heterozygote); at approximately 33% or approximately 66% for triploids (=ABB or AAB possible genotypes); and approximately 25%, 50% or 75% for tetraploids (=ABBB, AABB, AAAB genotypes, respectively). Log-likelihoods of the observed allelic depth distributions were extracted for Gaussian expectations under each fixed model (e.g. 2n, 3n or 4n), then normalized by that under a model of freely variable allele depths (=$\log L_{Fixed}/\log L_{Free}$), with chosen models representing the greatest normalized log-likelihood. Results were employed to generate prior expectations for subsequent ploidy-aware genotyping and downstream filtering.

Formatted SNPs (as .vcf) were genotyped for both *de novo* and transcriptome-guided assemblies using the Bayesian variant calling pipeline in POLYRAD [70]. We first computed per-locus $H_{IND}/H_E$ values, a statistic representing the probability of sampling reads from two different alleles at a given locus in an individual [71]. We then simulated expected $H_{IND}/H_E$ values given the sample size and read depth distribution from the observed data, under expectations of either diploidy or tetraploidy, to define threshold values under which markers are statistically consistent with Mendelian behaviour as a 95% upper-bound of the simulated distribution. Because expected values differ by ploidy, we used $H_{IND}/H_E$ values averaged across loci to segregate those statistically consistent with a diploid genotype model from those consistent with a tetraploid model. Using $H_{IND}/H_E$ thresholds and custom Python code (https://github.com/tkchafin/polyrad_scripts/filterPolyVCF.py), we partitioned genotypes for both assembled datasets into loci statistically consistent with tetraploid and diploid expectations. An additional requirement was that more than 50% of individuals be genotyped at an average mean per-individual depth of 20 X.

## 2.3. Geometric morphometrics of Nepali fishes

We first examined patterns of morphological convergence and divergence using geometric morphometric data [51] derived from 528 images captured from museum specimens representing all five Nepali species (for full sample metadata and KU voucher numbers, see Regmi *et al.* [51]; for those used for ddRAD, see electronic supplementary material, table S2). Our analyses were restricted to the Nepali sub-tree in that voucher specimens for Bhutanese samples were not available. Briefly, 18 landmarks were targeted, focusing on head morphology (i.e. snout elongation, head depth and eye/nostril placement), and body shape (origin/insertion of fins, upper/lower bases of the caudal fin and urostyle as most posterior point). Coordinates were then superimposed across samples using generalized Procrustes analysis implemented in GEOMORPH [80]. Full details on geometric morphometric data processing, including bias and sensitivity analysis, can be found in Regmi *et al.* [51].

Procrustes-aligned coordinates were summarized using a principal component analysis (PCA) in GEOMORPH [80]. We then conducted a linear discriminant analysis using the MASS R package [81], maximizing discriminant capacity with 80% of samples as a training set, and group classifications specified as species X drainage.

## 2.4. Phylogenetic relationships

Given that most phylogenomic methods cannot take into account polyploid SNP genotypes [82], our initial phylogenies were instead built using an alignment-free method that computes distances from the intersection of *k*-mer distributions in unaligned reads [83,84]. Matrices of *k*-mer distances were then used to infer a phylogenetic tree with branch lengths using FASTME [85].

The resulting topology was additionally contrasted with that inferred under a polymorphism-aware (POMO) model inferred in IQ-TREE [86,87], as evaluated across 1000 ultra-fast bootstraps, with a GTR substitution model, gamma-distributed rates ($N = 4$ categories) and a virtual population size of 19.

## 2.5. Population structure and molecular clustering

Most widely used clustering or 'assignment test' methods that examine population structure are either inappropriate for polyploid data or cannot handle mixed-ploidy genotypes [67]. However, STRUCTURE is robust in both situations [88,89], though more computationally intensive than alternatives. We thus

coded our mixed-ploidy genotypes as input for STRUCTURE, following recommendations of Meirmans *et al.* [67] and Stift *et al.* [89], using 10 replicates each of *K* values (= number of sub-populations) ranging from $K = 1$ to $K = 20$. Analyses were additionally replicated across *de novo* and transcriptome-guided assemblies, using a total MCMC length of 100 000 iterations following a 50 000-iteration burn-in. Input files were generated using *polyVCFtoStructure.py* (https://github.com/tkchafin/polyrad_scripts), and results were aggregated/visualized using the CLUMPAK pipeline [90], with the selection of optimal *K* following the Evanno method [91]. Final visualizations aligned the ancestry proportion bar plots aligned to our SKMER phylogeny using code from Martin *et al.* [92].

Results were contrasted with ordination via discriminant analysis of principal components (DAPC), performed in ADEGENET [93,94]. Samples were stratified according to species X basin assignment (e.g. *S. progastus* from Wang Chhu) and with analyses performed in three ways: (i) globally; (ii) Nepal-only; and (iii) Bhutan-only. The number of principal components (PCs) retained in each case was determined using the *xvalDapc* function as a cross-validation approach, with the optimal number of the root-mean-square-error of assignment derived from a 10% subset (with the remaining 90% serving as a training set). This was accomplished across 30 replicates per level of PC retention, up to a maximum of 300 retained PCs, resulting in 20 (globally), 10 (Bhutanese) and 5 (Nepali) retained PCs.

## 2.6. Modelling population mixture

We assessed hybridization within Nepali and Bhutanese sub-trees using TREEMIX [95], with a global search across numbers of migration events (*m*) ranging from 0 to 5. Because markers are assumed diploid, migration analyses were only performed on markers statistically fitting diploid expectations, as designated by POLYRAD [70]. Optimal values of *m* for each sub-tree were determined from rates of log-likelihood change as computed in OPTM [96], which generates an *ad hoc* statistic (ΔM) representing the second-order rate of change weighted by the standard deviation (e.g. the 'Evanno' method; Evanno *et al.* [91]). Independent replicates on the full dataset yielded identical likelihoods for some *m*-values (i.e. yielding an undefined ΔM) and, given this, we assessed variability using 100 bootstrap pseudoreplicates per migration model. To additionally discriminate among divergence scenarios among Nepali *S. richardsonii* and *S. progastus*, we calculated a 4-taxon Patterson's *D* statistic and admixture fractions ($f_4$-ratio) [97,98]. Both are formulated to test enrichment of shared-derived site patterns between either component of a lineage pair (P1 or P2), and a third lineage (P3) relative to an outgroup. The assumed phylogenetic structure would be: (((P1, P2), P3), P4). Our interest was in shared patterns, both at the level of 'conspecifics' among drainages and 'heterospecifics' within drainage, and our tests were conducted such that P1 and P2 were defined by river (e.g. (*rich*X, *prog*X), *rich*Y), Outgroup), where X and Y represent different drainages), and where P1 and P2 were defined per taxon-assignment (e.g. (*rich*X, *rich*Y), *prog*X), Outgroup)). The outgroup in all cases was *S. cf. oconnori* from Bhutan.

## 2.7. Testing models of co-divergence

To test if shared (e.g. geomorphic) events may have driven co-divergence, we used the program ECOEVOLITY [99], which employs a Bayesian method to compute probabilities on the number of independent divergences across a series of pairwise comparisons. Because ECOEVOLITY is most consistent when analysing both constant and polymorphic sites [99,100], we sampled full-locus alignments (excluding sites with either greater than 25% missing data globally or per-population) using the *phylip2ecoevolity.pl* script from Chafin *et al.* [101]. The event model followed a Dirichlet process, with the prior probability distribution for the number of divergence events skewed towards a model of complete independence (i.e. no co-divergence). Posterior probabilities were computed over a total of 75 000 MCMC iterations, with a sampling frequency every 50th iteration. Burn-in was automatically computed using an automated iterative procedure which selected the number of burn-in iterations which maximized the effective sample size.

## 2.8. Testing patterns of selection and locus-wise differentiation

Here, we sought loci strongly associating with axes of population differentiation and therefore employed the program PCADAPT [102] to identify loci putatively associated with differentiation of *S. richardsonii* and *S. progastus*. Detecting loci under selection is a persistent methodological gap in those studies examining polyploid SNP data [67] and, given this, we restricted our analysis to transcriptome-mapped loci that

could appropriately be genotyped as diploid [70]. Analyses were performed by partitioning the data so as to target specific divergence events: *S. richardsonii* X *S. progastus* (Gandaki); *S. richardsonii* X *S. progastus* (Koshi); *S. richardsonii* X Lake Rara (*S. macrophthalmus*, *S. raraensis* and *S. nepalensis*); *S. nepalensis* X *S. macrophthalmus* + *raraensis*; and *S. progastus* (Bhutan) X *S. cf. oconnori*. Loci were additionally restricted to those with a minor allele frequency greater than 0.05, with significance assessed using $\alpha = 0.01$ adjusted via Bonferroni correction (where $N$ tests = $N$ SNPs). Transcripts encapsulating outlier SNPs were additionally annotated with gene ontology (GO) terms for biological functions using the BLAST2GO pipeline [103], searching against the SWISS-PROT and InterPro databases [104,105]. Redundancy was reduced in GO-term annotations by measuring semantic similarity using RRVGO [106] and the *Danio rerio* organismal database.

We additionally contrasted population pairs (delineated as above), using a ploidy-appropriate $F_{ST}$ measure. We calculated heterozygosity (i.e. 'gene' diversity [92]) as: $H_s = 1 - \Sigma p_i^2$, where $p_i$ is the frequency of allele $i$ [107]. Although in diploids, this measure is often referred to as 'expected heterozygosity', it does not have the same relationship with heterozygosity across ploidies [67]. Finally, we calculated genetic distance as Jost's $D$ [108], as well as absolute distance ($D_{XY}$) for each SNP based on allele frequencies and summarized across loci as an arithmetic mean [109].

## 2.9. Partitioned analysis across subsets of loci

As a final test of the hypothesis that divergence between pairs of *S. richardsonii*–*S. progastus* represents isolation prior to secondary hybridization, we performed a partitioned analysis across subsets of our data. Our prediction was as follows: if differential adaptation reflects a retention of minor genomic regions from secondary introgression (i.e. as opposed to arising independently), then disparate populations of *S. richardsonii* would be more similar at loci for which drainage pairs (*richardsonii*–*progastus*) are highly diverged. We diagnosed such loci in two ways: first, we computed a ratio of $D_{XY}$ between *richardsonii*–*progastus* pairs within Koshi and Gandaki rivers, and $D_{XY}$ of *S. richardsonii* populations among drainages. Here, a value less than 1.0 indicates a locus for which *richardsonii*–*progastus* pairs are more similar than *richardsonii*–*richardsonii* comparisons. A value greater than 1.0 indicates a locus for which *richardsonii-progastus* are more highly diverged. If loci with a $D_{XY}$ ratio greater than 1.0 originated from hybridization, genetic relationships among those loci should reflect a conspecific phylogeny. To assess the latter, we replicated TREEMIX analyses (as above) for loci greater than 1.0 and less than 1.0. As a secondary test, we also partitioned locus-wise *richardsonii*–*progastus* $F_{ST}$ into four groups, each receiving the same analytical treatment as the $D_{XY}$ ratio partitions: (i) $F_{ST}$ = 0.0–0.249; (ii) 0.25–0.49; (iii) 0.50–0.749 and (iv) 0.75–1.0.

# 3. Results

## 3.1. Sequence assembly and genotyping

Prior to downstream filtering, we assembled $N = 48\,350$ and $14\,301$ SNPs for *de novo* and transcriptome-guided assemblies, respectively (electronic supplementary material, table S3). Allele depths for biallelic sites were overwhelmingly trimodal, with NQUIRE likelihoods suggesting tetraploidy across all study taxa (figure 3). Three samples failed to yield genotype data for greater than or equal to 50% of loci and were removed. Simulated expectations in POLYRAD yielded $H_{IND}/H_E$ thresholds of 0.52 (diploid) and 0.65 (tetraploid). Post-filtering datasets consisted of 5601 (2n) and 5284 (4n) transcriptome-mapped, and 15 547 (2n) and 19 772 *de novo* SNPs (electronic supplementary material, table S3). For each assembly method, these were divided into files containing all loci as well as subsets containing only those statistically consistent with a diploid model. Scripts for formatting datafiles and outputs can be found at: https://github.com/tkchafin/polyrad_scripts.

## 3.2. Contrasting molecular and geometric morphometric results

Morphological DAPC showed a marked convergence of *S. richardsonii* and *S. progastus* body shapes, regardless of origin, with Lake Rara taxa (*S. macrophthalmus*, *S. nepalensis* and *S. raraensis*) having weak or no differentiation (figure 4). Among the latter, *S. nepalensis* was most intermediate between a 'Lake Rara' cluster, and that for *S. richardsonii* (and to a lesser extent, *S. progastus*). This contrasted sharply with the clustering of Nepali specimens based solely on ddRAD data (figure 5a,c), where the

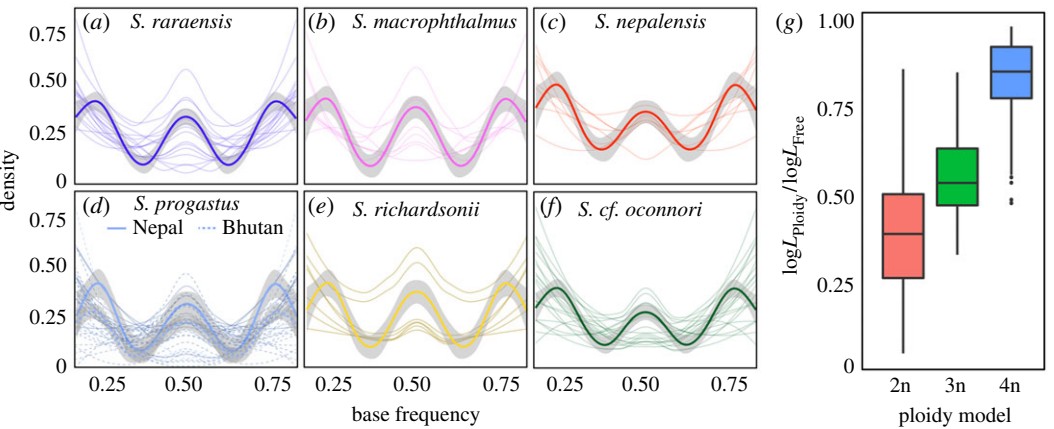

**Figure 3.** Evaluation of ploidy from $N = 48\,350$ ddRAD loci for six *Schizothorax* species. (*a*–*f*) Distributions of base frequencies among raw reads representing bi-allelic SNPs for six *Schizothorax* spp. collected from Nepal (*S. macrophthalmus* (*b*), *S. nepalensis* (*c*), *S. progastus* (solid line) (*d*), *S. raraensis* (*a*) and *S. richardsonii* (*e*)) and Bhutan (*S. cf. oconnori* (*f*) and *S. progastus* (dashed line) (*d*)). (*g*) Likelihoods from Gaussian mixture models of fixed ploidy (e.g. diploid, triploid and tetraploid), divided by the likelihood under a model of freely variable base frequencies (=$\log L_{Ploidy}/\log L_{Free}$).

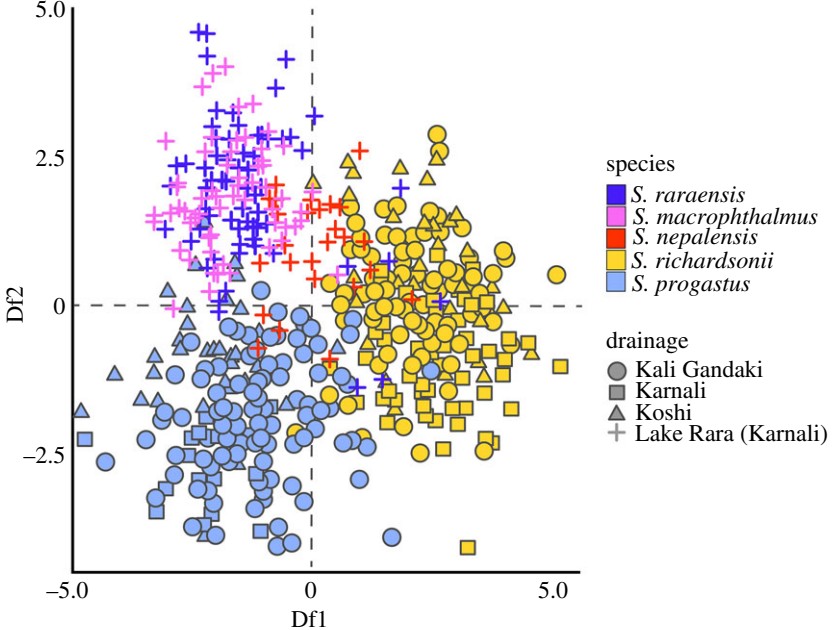

**Figure 4.** DAPC of morphological shape variation among five *Schizothorax* species. Results represent $N = 528$ individuals of *S. macrophthalmus*, *S. nepalensis*, *S. progastus*, *S. raraensis*, and *S. richardsonii* collected from the Gandaki, Karnali (to include Lake Rara) and Koshi drainages of Nepal, and Procrustes-aligned two-dimensional Cartesian coordinates for 18 anatomical landmarks.

predominant relationship was by drainage (e.g. with Koshi and Gandaki *S. progastus* and *S. richardsonii* grouping together).

The latter was mirrored in our phylogenetic analyses, with pairs of *S. progastus–richardsonii* from the Koshi and Gandaki rivers being monophyletic in the alignment-free phylogeny (figure 6*a*) and agreeing approximately with STRUCTURE-inferred assignment probabilities (figure 6*b*). Optimal Evanno-derived *K* in the latter identified peaks in *DeltaK* at *K* = 3 and *K* = 7. The former yielded a homogeneous Nepali sub-tree, with only the Koshi River samples somewhat distinct (figure 6*a,b*), whereas *K* = 7 reveals weak differentiation, with mixed assignment spanning basins.

Inferred edges in TREEMIX suggested migration between *S. progastus* of the Koshi and Gandaki rivers, as well as between *S. progastus* of the Koshi with *S. raraensis* and *S. macrophthalmus* of Lake Rara (figure 6*c*). However, the latter may be mis-ascribed, particularly given the absence of sampling for *S. progastus* from the Karnali River. Furthermore, migration edges in TREEMIX may also explain the

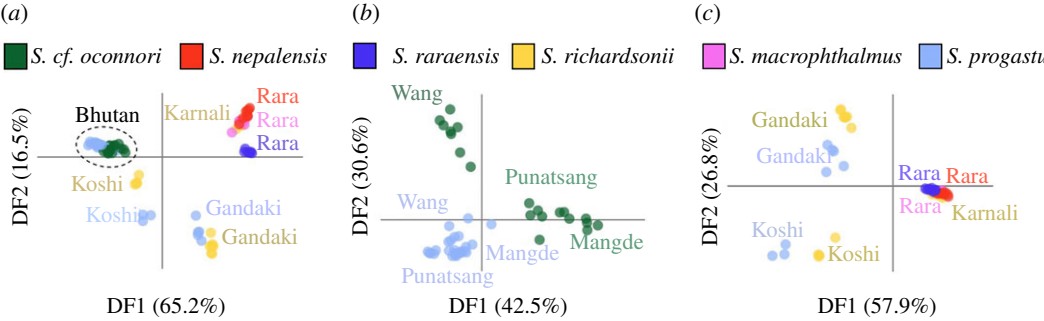

**Figure 5.** Visualization of genetic variation among six *Schizothorax* species, as assessed across $N = 15\,547$ statistically diploid SNPs. Visualizations represent the first two discriminant functions from a DAPC-computed across (*a*) all samples from Bhutan (*S. cf. oconnori* and *S. progastus*) and Nepal (*S. macrophthalmus*, *S. nepalensis*, *S. progastus*, *S. raraensis* and *S. richardsonii*); (*b*) only those samples from Bhutan, additionally partitioned by major drainage encompassing the samples locality (i.e. Wang Chhu, Mangde Chhu or Punatsang Chhu) and (*c*) only individuals samples in Nepal. Nepali samples are partitioned by major drainage (i.e. Gandaki, Koshi and Karnali), with species from Lake Rara (*S. macrophthalmus*, *S. nepalensis* and *S. raraensis*) additionally separated.

mixed inter-drainage assignment seen in Sᴛʀᴜᴄᴛᴜʀᴇ, as well as the non-monophyly of Koshi River *richardsonii–progastus* in our PoMo analysis (figure 7).

Bhutanese *Schizothorax* spp. differed in that no clear delineations were found between high- and low-elevation clades in *S. progastus*, and with clustering instead reflecting species-level assignments (figures 5 and 6*b*). Sᴛʀᴜᴄᴛᴜʀᴇ results suggested mixed assignment of *S. progastus* and *S. cf. oconnori*, particularly in the Wang Chhu where both *de novo* and transcriptome-guided results agreed (figure 6*b*). TʀᴇᴇMɪx results also revealed evidence in multiple sub-basins for exchange across species boundaries (figure 6*c*; electronic supplementary material, S1). Of note, mixed assignment was also observed between *S. progastus* (Bhutan) with those from Nepal (figure 6*c*). We could not test whether this reflects retained ancestral variation (i.e. prior to divergence of Ganges and Brahmaputra rivers), or more contemporary mixture. Patterson's *D*-statistic likewise supported this mixed ancestry, both at the level of conspecifics among drainages and heterospecifics within drainages (electronic supplementary material, table S3).

## 3.3. Co-divergence analysis

Co-divergence analysis rejected co-divergence for all riverine pairwise comparisons for both Nepal and Bhutan (figure 8). One noTable exception was the inferred simultaneous divergence of the 'Lake Rara' lacustrine radiation (here represented as two comparisons: *S. macrophthalmus* x *S. raraensis* + *S. nepalensis*; and *S. raraensis* x *S. nepalensis*; figure 8*a*). Here, a model of four divergences was selected for Nepal (posterior probability greater than 0.9; figure 8*c*), with the relative timing for this radiation exceptionally recent compared with those inferred for within-drainage comparisons. We note, however, that inferred divergence times are probably skewed by introgression (e.g. Figure 6*c*), but that the exact nature of this effect is unclear [101]. Parameter estimates for all analyses post burn-in were greater than 600.

## 3.4. Locus-wise differentiation and outlier analysis

The greatest number of significant SNPs following Bonferroni correction occurred in outlier analysis involving Koshi and Gandaki *S. richardsonii–progastus* pairs, and from the comparison of *S. progastus* and *S. cf. oconnori* from Bhutan (i.e. $N = 531$ and 54, respectively; electronic supplementary material, figure S2*a*). Of these, the most substantial number of overlapping SNPs was found among Koshi and Gandaki comparisons. Outliers in both cases had higher $F_{ST}$ than did the genome-wide distribution involving *de novo* or transcriptome-guided assemblies. This pattern was not repeated in the same loci for Bhutan (electronic supplementary material, figure S2*b*), although outlier loci in Bhutan did share a negative relationship of gene diversity ($H_E$), as compared with *S. richardsonii–progastus* $F_{ST}$ (electronic supplementary material, figure S2*c*). Outlier $F_{ST}$ for *S. richardsonii–progastus* was correlated in the Gandaki versus Koshi rivers, but with no discernible relationship with the same measure when compared with Karnali River *S. richardsonii* and Lake Rara species, Bhutanese *S. progastus–oconnori*, nor among Bhutanese populations of only *S. progastus* (electronic supplementary material, figure S3). Relationships among different locus-differentiation statistics otherwise occurred per theoretical expectations (electronic supplementary material,

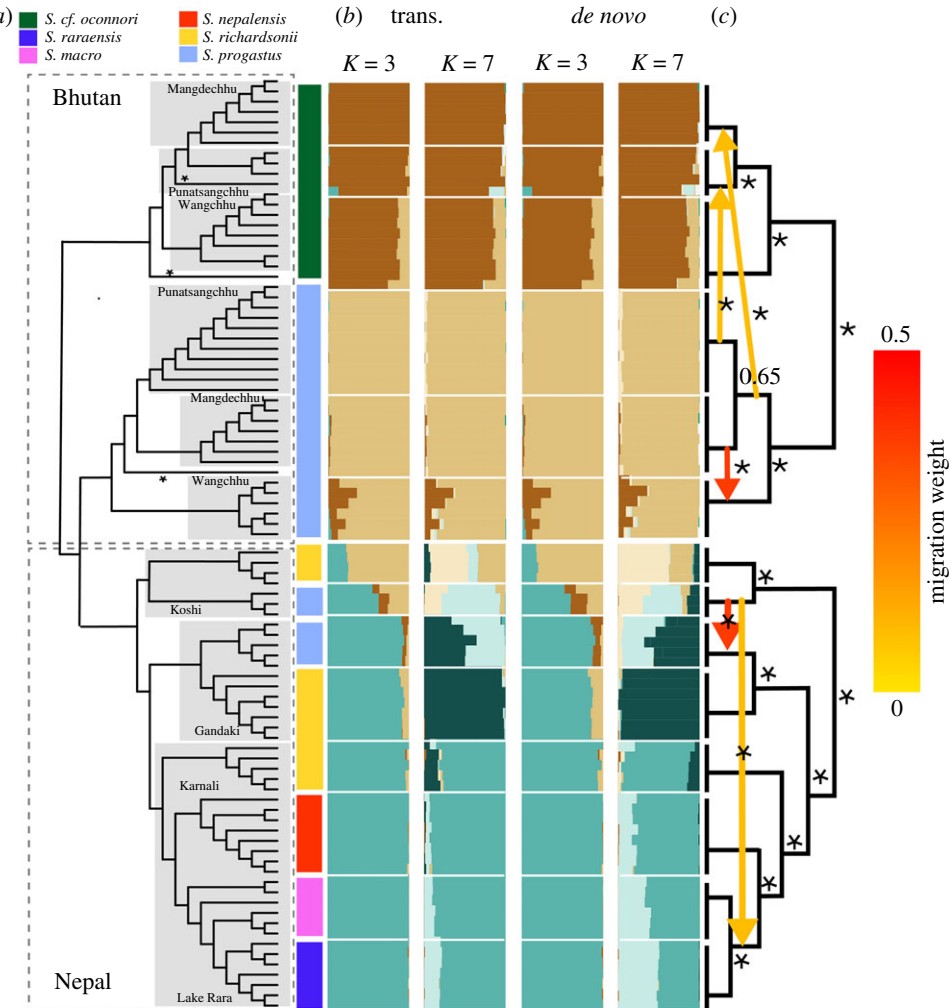

**Figure 6.** Phylogenetic and population structure of $N = 93$ *Schizothorax* spp. collected in major drainages of Bhutan and Nepal. Results represent (*a*) an unrooted alignment/assembly free phylogeny inferred from *k*-mer distances, computed from raw ddRAD reads. Clades are grouped by drainage, and with coloured bars denoting tips grouped by taxon (*S. macrophthalmus*, *S. nepalensis*, *S. cf. oconnori*, *S. progastus*, *S. raraensis* and *S. richardsonii*). (*b*) Bar plots are provided for population assignments at $K = 3$ and $K = 7$ for (left) 10 884 transcriptome-aligned SNPs and (right) 35 319 *de novo* assembled SNPs. (*c*) TREEMIX analyses of population mixtures, with coloured arrows representing inferred migration weights.

figure S4). Hence, only comparisons (above) are presented for $F_{ST}$. Relating $D_{XY}$ ratios to $D_{XO}$ showed that high values were probably driven by near-zero distances among *S. richardsonii* pairs for a subset of loci (electronic supplementary material, figure S5), in line with evidence from the *D*-statistics indicating shared inter-drainage ancestry among conspecifics (electronic supplementary material, table S3).

Although approximately 75% of outlier transcripts could not be assigned candidate protein products, GO term enrichment for shared outliers revealed a number of related biological processes, including anatomical structure formation (GO: 0048646), upregulation of phosphoprotein phosphatase activity (GO: 0032516) and upregulation of G protein-coupled receptor signalling (GO: 0045745) (electronic supplementary material, figure S2*d*). The few outlier transcripts that could be assigned candidate proteins in the overlapping Gandaki-Koshi set were Aurora Kinase A (*AURKA*), Zinc Finger SWIM domain containing protein 6 (*ZSWIM6*), Adhesion G protein-coupled receptor A3 (*ADGRA3*) and Caspase B (*CASPB*).

## 3.5. Partitioned ancestry and mixture analyses

Loci for which *richardsonii–richardsonii* divergence was greater than *richardsonii–progastus* (i.e. $D_{XY}$ ratio less than 1.0; figure 9*a*) occurred in drainages irrespective of ecomorphological taxon-assignment and corresponded to approximately 68%. The remaining approximately 32% yielded groupings consistent with

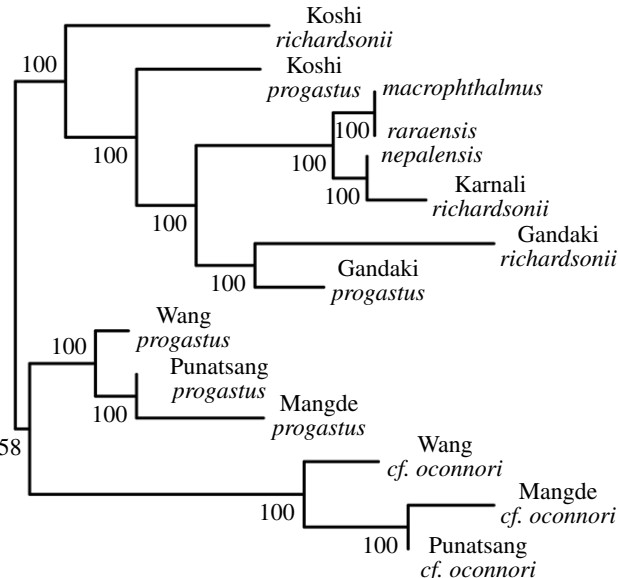

**Figure 7.** Population tree generated for $N = 93$ *Schizothorax* spp. divided into 13 populations using a polymorphism-aware model applied to 15 543 ddRAD SNPs. Nodal values represent bootstrap support expressed as percentage of 1000 ultra-fast bootstraps.

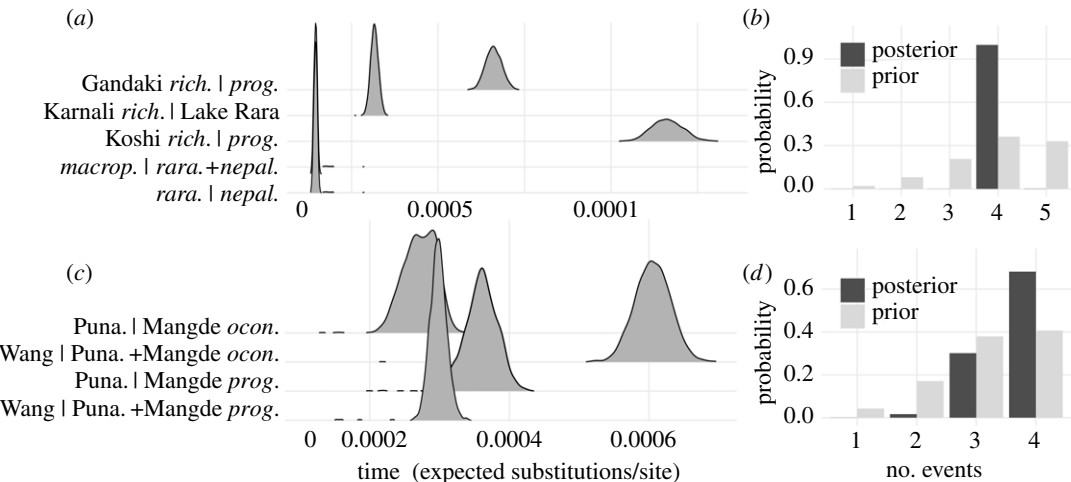

**Figure 8.** Tests of co-divergence among lineage pairs in Nepal and Bhutan; (*a*) and (*c*) show posterior probability distributions on divergence times for Nepal and Bhutan, respectively. Nepalese results (*a*) represent divergence of *Schizothorax richardsonii* (=rich.) and *S. progastus* (=prog.) in the Gandaki and Koshi rivers, *S. richardsonii* from the Karnali River with the 'Lake Rara' complex, *S. macrophthalmus* (=macrop.) with *S. raraensis* (=rara.) and *S. nepalensis* (=nepal.). Results for Bhutan (*c*) depict *S. cf. oconnori* (=ocon.) and *S. progastus* from the Punatsang Chhu (=Puna.), Mangde Chhu and Wang Chhu rivers. (*b,d*) Posterior (dark bars) and prior (light bars) probabilities of the numbers of co-divergences.

species-level taxonomy, but with migration among species suggested within the Gandaki and Koshi rivers (figure 9*a*). Repeating the same analysis across loci binned by $F_{ST}$ yielded a transition from drainage level at low values (with migration among mid-highland *S. progastus*) towards species level at high values (with migration within drainages) (figure 9*b*). This pattern was consistent when loci were binned with $F_{ST}$ estimates derived either from *S. richardsonii–progastus* comparisons in the Gandaki or the Koshi, although the transition towards species-level grouping was slightly protracted in the latter (figure 9*b*).

## 4. Discussion

We applied genome-wide SNP data and ploidy-aware genotyping to demonstrate that genetic and morphologically distinct *Schizothorax* forms in Himalayan tributaries evolved prior to secondary gene

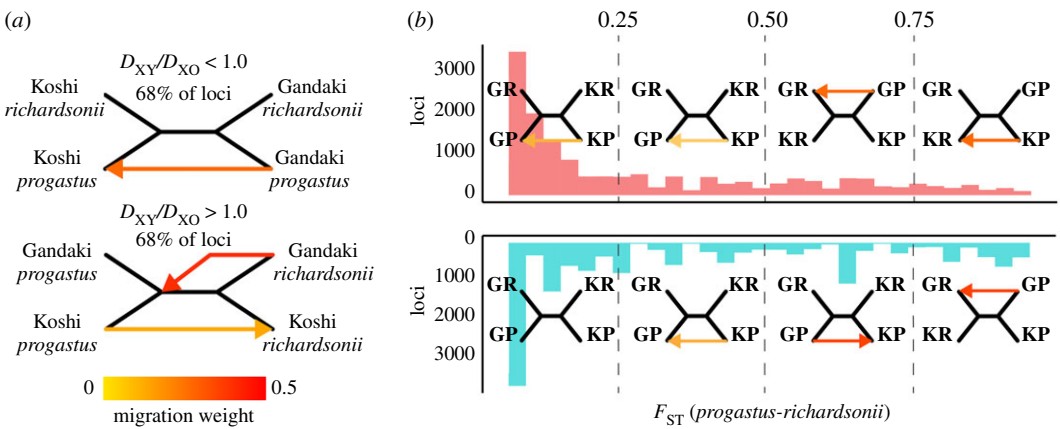

**Figure 9.** TREEMIX analyses of population mixtures among S. progastus and S. richardsonii pairs collected from the Gandaki and Koshi rivers of Nepal, based on partitioning different subsets of loci. (a) Loci were grouped by ratios of $D_{XY}$ (computed between heterospecifics within each drainage) and $D_{XO}$ (computed between conspecific S. richardsonii), with those loci having a $D_{XY}$ ratio greater than 1.0 in both Gandaki and Koshi comparisons (i.e. greater distances among S. richardsonii and S. progastus within drainage than S. richardsonii among drainages) analysed separately. (b) Loci were also partitioned into four bins defined by per-locus $F_{ST}$ computed in Gandaki (above; red) and Koshi (below; blue) pairs. Coloured arrows indicate inferred migrations within each partitioned analysis, with the colour indicating the migration weight.

flow (i.e. Scenario (c); figure 1), rather than as a parallel emergence of elevational forms. In addition, highland specialists in Nepal converged strongly onto a singular phenotype that was recapitulated in phylogenetic patterns at presumed targets of selection. However, neutral variation showed signs within each basin of homogenization among elevational pairs (S. richardsonii and S. progastus), suggesting a breakdown of reproductive isolation. In Bhutan, elevational pairs (S. progastus and S. cf. oconnori) exhibited low-level admixture, suggesting that elevational pairs possess either greater reproductive fidelity or instead represent an earlier stage of homogenization via secondary gene flow.

## 4.1. Parallel divergence versus parallel hybridization in Nepali snowtrout

Genome-wide relationships among Nepali and Bhutanese *Schizothorax* give the appearance of recurrently emerging ecotypes, with repeated colonization of highland habitats from a mid-highland progenitor occurring within each drainage (figures 5 and 6). This seemingly fits the narrative of whole-genome duplication as promoting the adaptive potential of schizothoracine fishes, as well as the broader pattern of convergence among high-elevation specialists [50,62]. Indeed, polyploid adaptive potential may have played a role in the parallel colonization of highland habitats by *S. richardsonii* in the Ganges tributaries of Nepal, and *S. cf. oconnori* in the Brahmaputra tributaries of Bhutan, although we note that the current study lacks geometric morphometric data for the latter.

In the case of elevational *Schizothorax* pairs in Nepal, we found that dominant ancestries shifted from grouping at the drainage level (based on scarcely differentiated/'neutral' loci) (irrespective of eco-morphology) towards the conspecific level (via strongly differentiated/selected loci). However, this raises an additional question: how are adaptive phenotypes retained despite introgression being reflected within most of the genome? Here, one potential is that an allopolyploid origin of *Schizothorax* conflates the signal of hybridization. However, a lack of sub-genome divergence was found in *S. oconnori* [110], as would be expected under allopolyploidy.

It is also possible that polyploidy facilitates a greater degree of genomic exchange when compared with reproductively isolated diploid species [111]. For example, ploidy has been suggested to promote introgression in diploid–tetraploid crosses of *Arabidopsis*, due to a circumvention of dosage-mediated postzygotic isolation [112–114]. Increased introgression has also been supported among tetraploid–tetraploid crosses (as herein), with an increase in local recombination rates as one potential mechanism. As ploidy increases, there is a relaxation of linkage as a component of purifying selection [115,116]. Similarly, increased dosage may 'mask' deleterious loads, especially in young polyploid species [117,118]. Given the young age of whole-genome duplication estimated for *S. oconnori* [110], these predictions implicate a genomic landscape vastly more porous than might be expected in their

diploid counterparts [114]. This may also be an underappreciated mechanism promoting the hypothesized increased adaptability of polyploids to stressful or novel environments [119,120].

## 4.2. Varying rates of hybridization among drainages

In contrast with the substantially permeable genomic landscape seen among *S. richardsonii* and *S. progastus* pairs in Nepal, we saw a much smaller signature of gene flow in Bhutanese elevational pairs (*S. progastus*, *S. cf. oconnori*). In addition to gene flow obscuring species boundaries [92,121], contrasts between Himalayan streams and extreme freshwater habitats in the American Southwest [121–123] also implicate the potential for anthropogenically mediated hybridization. Here, 'extinction vortices' may be driven by a coupling of potentially maladaptive hybridization combined with declining population sizes. This provides a backdrop against which climatic expectations of shrinking habitats for vulnerable highland species such as *S. richardsonii* can be contrasted [124].

The potential role of hybridization in demographic trends or extinction risks [125,126] cannot be inferred from our data, nor can the environmental covariates potentially modulating species boundaries be inferred without further work. Given the potential for seasonal migratory behaviour in *Schizothorax* [73,127], hydropower dams will disrupt movements, thus inadvertently promoting interspecific contact. Here, we again emphasize parallels with the heavily modified and regulated rivers of western North America, where water policy and impoundments promote hybridization [122,128], define habitat suitability [129] and alter environmental cues [130]. Though both countries have a high percentage of protected areas, our results suggest that anthropogenically mediated hybridization represents an additional dimension to consider when balancing freshwater conservation planning [131,132].

## 4.3. High-elevation adaptation

Although we found relatively little overlap in putative targets of selection (electronic supplementary material, figure S2*a*), our reduced-representation approach only surveyed a small percentage of transcripts, with poor success in establishing functional annotations. Molecular convergence among high-elevation populations has been observed in disparate human populations [133], and even among humans and their domesticated species [134]. Other studies in *Schizothorax* have shown an array of candidate genes underscoring adaptations to hypoxic and ultraviolet conditions associated with elevation [135].

Schizothoracine fishes in general have been characterized as species similarly to trout (Salmonidae) adapted to torrential flows, with positioning in rapid currents facilitated by body shape [136]. A longitudinal survey of Nepali assemblages showed the replacement of *S. richardsonii* by *S. progastus* at decreasing elevations [60], suggesting a role for factors such as dissolved oxygen and flow rates [137]. Turnover and narrow elevational ranges are found in other Himalayan taxa as well [138,139]. Specialized sucker-like adaptations are present in *S. richardsonii* but absent in *S. progastus* [60], and species richness in *Schizothorax* is greater at mid-elevation [140], seemingly corroborating this relationship in *Schizothorax*.

An interesting case of adaptation that we did not explore fully herein is that of the putative species flock in Lake Rara (figure 8*a*). There, species overlap in geometric morphometric space (figure 3), yet display marked differences in mouth morphology, gill raker shape, spawning microhabitat choice and diet (with *S. raraensis* insectivorous, *S. nepalensis* herbivorous and *S. macrophthalmus* planktivorous) [53]. We found them genetically indistinguishable across several analyses, in agreement with previous studies [54,56], though this is unsurprising given their apparent recent and rapid diversification (figure 8) [101].

## 5. Conclusion

Our results indicate that population genomics can be accomplished in a statistically appropriate manner using non-model polyploid species with relatively minor adjustments to the traditional molecular ecology workflow. However, several limitations were encountered (such as the lack of ploidy-aware methods for outlier detection), signalling a need for further development. Despite these limitations, we were able to disentangle the parallel evolution of *Schizothorax* species in Himalayan tributaries of the Ganges and Brahmaputra rivers, and with recurrent processes implicated at differing timescales: convergent adaptation towards high-elevation environments among major drainages (e.g. Ganges versus Brahmaputra), and parallel selection against introgressive homogenization within drainages.

With regard to the latter, we found heterogeneous levels of admixture among populations. Finally, we rejected the hypothesis of co-divergence between highland and mid-highland riverine forms, yet found the lacustrine radiation in Lake Rara to be both recent and rapid.

Ethics. All methods were performed in accordance with relevant guidelines and regulations. Bhutanese collecting permits were in conjunction with the National Research & Development Centre for Riverine and Lake Fisheries (NRDCR&LF), Ministry of Agriculture & Forests (MoAF), Royal Government of Bhutan. The export of fin clips was authorized through a Material Transfer Agreement (MTA) provided by the National Biodiversity Centre (NBC), with additional approvals provided by the Department of Forests & Park Services (DoFPs) and Bhutan Agricultural and Food Regulatory Authority (BAFRA). Sampling protocols were approved by the University of Arkansas Institutional Animal Care and Use Committee (UA_IACUC_17064). The study was also carried out in compliance with ARRIVE guidelines (https://arriveguidelines.org).

Data accessibility. Raw sequence data is available via the NCBI Short Read Archive (SRA) under BioProject PRJNA759907. Assembled datasets and input files are available in the Open Science Framework repository (doi:10.17605/OSF.IO/TMJFK). Relevant code for this research work are stored in GitHub: https://github.com/tkchafin/polyrad_scripts and https://github.com/tkchafin/scripts, and have been archived within the Zenodo repositories doi:10.5281/zenodo.5393418 and doi:10.5281/zenodo.5181290.

Authors' contributions. All contributed to conceptualization and study design. D.R.E. collected samples in Nepal. S.T., K.W., S.D., P.N., S.N., C.C. and G.P.K. coordinated and supervised fieldwork in Bhutan. S.T., K.W., M.E.D., M.R.D, S.D., P.N., S.N., C.C., G.P.K., and T.K.C. collected samples in Bhutan. B.R. collected and curated geometric morphometric data. M.E.D. and M.R.D. generated molecular data. T.K.C. wrote necessary codes and performed bioinformatic work. B.R. and T.K.C. analysed data. T.K.C. generated figures and drafted the manuscript. All authors contributed to revising the final product and subsequently approved its submission.

Competing interests. We declare we have no competing interests.

Funding. The National Research Centre for Riverine and Lake Fisheries (NRCRLF, Bhutan) provided logistic support and personnel with which to sample fishes. Analytical resources were provided by the Arkansas Economic Development Commission (Arkansas Settlement Proceeds Act of 2000) and the Arkansas High Performance Computing Center (AHPCC). Computational support was also provided by the U.S. National Science Foundation (NSF) funded XSEDE Jetstream cloud (award no. TG-BIO200074 to T.K.C.). This research was made possible through generous endowments to the University of Arkansas: The Bruker Professorship in Life Sciences (M.R.D.), the Twenty-first Century Chair in Global Change Biology (M.E.D.), Distinguished Doctoral Fellowship award (T.K.C.) and Graduate Teaching Assistantships (K.W., T.K.C.), all of which provided salaries and/or research funds to complete this study. T.K.C. received additional support via an NSF Postdoctoral Fellowship in Biology under grant no. DBI: 2010774. Any opinions, findings, and conclusions or recommendations expressed in this material are those of the author(s) and do not necessarily reflect the views of the funding agencies nor affiliated organizations.

Acknowledgements. Museum specimens and tissue samples for Nepal were obtained from the University of Kansas Natural History Museum (KUNHM; https://biodiversity.ku.edu/ichthyology/collections), vouchers KU:KUIT:27806–27815, 27885–27887, 29043, 29050, 29233, 29234 and 29236–29238 (for specific tissue numbers, see electronic supplementary material, table S2). We thank Curator Leo Smith and Collection Manager Andrew Bentley for tissue loans and use of imaging facilities at KUNHM. Nepali samples were originally collected by D.R.E., as funded by a Fulbright Scholarship, the Explorers Club, and the National Geographic Society (NGEO) Committee for Research.

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
