## [Peer Review File · Royal Society Open Science]

Review History

RSOS-210727.R0 (Original submission)

Review form: Reviewer 1

Is the manuscript scientifically sound in its present form?

Yes

Are the interpretations and conclusions justified by the results?

Yes

Is the language acceptable?

Yes

Do you have any ethical concerns with this paper?

No

Have you any concerns about statistical analyses in this paper?

No

Recommendation?

Accept with minor revision (please list in comments)

Comments to the Author(s)

This study did a population genetics in polyploid snowtrout in Schizothorax with emphasis of species *S. progastus* and *S. richardsonii*. I found that the study in general is interesting and well conducted. I think that the main finding is that the authors found that *S. progastus* and *S. richardsonii* from Grandaki and Koshi genetically close to each other, which results from gene flow of secondary contact. However, I have the following concerns before its publication. First, I do not feel that analyses and results in the study really touch its topic of parallel adaptations to an altitudinal gradient in polyploid Schizothorax species. The results in Figure 4 and 5 do not completely reflect parallel genetic variation and the selection analysis does not touch either parallel genetic variation or parallel adaptations to an altitudinal gradient. Those results and analyses are only valid to gene flow of secondary contact between *S. progastus* and *S. richardsonii*. Second, since the study is working on polyploid species, it is important to consider read coverage when calling SNP. However, read coverage is not justified. It seems the authors used "6" that is too low for SNP calling in polyploid species. In addition, I have no idea what "transcriptome-guided assemblies" is. Third, there are several studies tried to work on population genetics of polyploid species, as the cited Meirmans et al. (2018). I am wondering how the results would be if analyses restrict to those tetraploid SNPs in Figure 2.

Review form: Reviewer 2**Is the manuscript scientifically sound in its present form?**

Yes

Are the interpretations and conclusions justified by the results?

Yes

Is the language acceptable?

Yes

Do you have any ethical concerns with this paper?

No

Have you any concerns about statistical analyses in this paper?

No

Recommendation?

Accept with minor revision (please list in comments)

Comments to the Author(s)

Review of Chafin_etal_ParallelAdaptationHimalayanFishes_RSOS

This paper employs a reduced representation library (modified ddRAD) approach to collect genome wide distributed SNP data for populations of Himalayan fishes to investigate parallel evolution. The paper then attempts to disentangle alternative hypotheses about the parallel evolution of adaptive ecotypes from either independent mutational events or a single shared

mutation that subsequently spreads through gene flow followed by adaptive increases in allele frequency.

This is a complex system, with multiple nominal morphological species defined, as well as ancestral polyploidy complicating genetic inference. While generally well written, I found some sections to be difficult to understand. For example, the description of the alternative hypotheses and related expectations for the resulting genetic data described in the Introduction to be unclear. This seems further complicated by the models of how adaptive variation may spread through within-species metapopulation dynamics versus a model of speciation through adaptation. The former is more interactive, whereas the latter sort of presumes a directional movement towards speciation. I urge the authors to work on the clarity of the writing and to use consistent language throughout the text.

General Issues:

Poidy; The ancestral polyploidy of these species complicates the analysis, and the paper describes a 'ploidy-aware' genotyping pipeline. The description appears clear; however, I have not used this pipeline and did not validate their approach myself. More generally, polyploidy and rediploidization is a complex issue but the abstract states that they "unambiguously quantified ploidy levels...". Given the complexities I think this should be tempered somewhat.

>>Language, e.g. use of 'parallel' line 58, and 'independent' on Line 478. This is a crux point for understanding your interpretation, since parallel evolution from standing genetic variation could still be considered "independent" evolution of the phenotype but based on the same genetic variant rather than an independent new mutation. Please check your use of this language throughout the manuscript.

Finally I have found it impossible to review this manuscript without exposing my bias towards salmonid research and suggesting citations to my own papers, so will sign the review. I would be happy to clarify or discuss any aspects of this review, or to comment on a revised version of the manuscript. – Devon Pearse.

Line comments:

Abstract line 12; language "When genotyped SNPs were clustering...". The next sentence is also strangely worded-- if I understand correctly it is essentially describing an outlier analysis, but I don't think I understand the final part about genomic homogenization.

Intro:

Line 35: Here it is unclear what the contrasting hypotheses are-- What is the expectation of the relative importance of factors that might cause "genomic islands"?

>I can't follow the logic being described in the three paragraphs from Line 38-59. How do the alternative 'hypotheses' described lead to different predictions about patterns in the genetic data? Also reference 29 Edelaar et al. 2008 seems like an old and not very relevant reference to cite and I am surprised by lack of references to much newer highly relevant work on parallel adaptive evolution and maintenance of 'genomic islands' especially in Salmonids, e.g. Larson et al. 2017; Pearse et al. 2014; 2019) as well as other fish taxa such as sticklebacks (Eda1; Jones et al. 2012) and Cod (genomic islands) as well as birds (e.g Ruegg et al. 2014 Mol Ecol) and related discussion in the literature. Would be good to connect this paper to broader literature. Finally Line 59 ends by referring to the 'speciation process', but the processes being described really seem intraspecific in

nature, occurring in metapopulations distributed in patchy habitat with alternative adaptive ecotypes.

Line 116; Sorry for even more self-promotion, but for comparison Campbell et al. 2019, G3, examined diploid and tetraploid ohnologs from a much older whole genome duplication in salmonids.

Line 126; Here I am not sure what "...deletion of historic ancestry..." means. Please clarify your language.

Methods

Line 300, Section (h): Dividing loci >1 or <1 for the ratio of Dxy divergence – seems like loci close to 1 really have no signal on which to base placing them in one group or the other. Should there be a gap in the thresholds above and below which they are partitioned?

Results

Figure 4b shows Bhutan genetic results, the widespread low elevation form is uniform. This result suggests that there is ongoing higher geneflow among populations of *S. prognathus* in the different watersheds, but less movement among *S. richarsonii* populations. This is very comparable to what happens in isolated trout populations, e.g. Pearse et al. 2014

Lines 370, 372; what are these numbers and substantial numbers? This is hard to infer from the figure, except for the 31 overlapping loci.

Discussion

Line 432; This is a clear statement of the authors' interpretation of the results and I agree with the way it is stated here except for the phrase 'subsequent homogenization...'. How was this shown to be true as opposed to alternative situations such as ongoing gene flow maintaining homogenization genome-wide except at adaptively important loci? Also Figure 7 doesn't appear to exist.

Line 435; 'rampant introgression', even a relatively small amount of ongoing gene flow will prevent genome wide divergence and maintain F_{st} near zero despite the presence of ecotypes maintained by specific adaptive genetic loci. See recent paper on run-timing ecotypes in salmon (Thompson et al. 2020 Science).

Lines 478-525; This section seems long and speculative and mostly repeats earlier conclusions. I suggest editing it down to just one or two paragraphs.

Decision letter (RSOS-210727.R0)

Dear Dr Chafin

The Editors assigned to your paper RSOS-210727 "Parallel adaptations to an altitudinal gradient persist in tetraploid snowtrout, despite extensive genomic exchange within adjacent Himalayan rivers" have now received comments from reviewers and would like you to revise the paper in accordance with the reviewer comments and any comments from the Editors. Please note this decision does not guarantee eventual acceptance.

Please submit your revised manuscript and required files (see below) no later than 21 days from today's (ie 23-Jun-2021) date. Note: the ScholarOne system will 'lock' if submission of the revision is attempted 21 or more days after the deadline. If you do not think you will be able to meet this deadline please contact the editorial office immediately.

on behalf of Dr Joachim Mergeay (Associate Editor) and Kevin Padian (Subject Editor)
openscience@royalsociety.org

Editor comment:

Thanks for your submission. Although the reviewers recommend "accept with minor revision," in our experience their concerns merit "major revision" so that you can have the time to consider and respond to their issues. To this end we wish you the best, and please address their thoughts fully and individually. Thanks.

Associate Editor Comments to Author (Dr Joachim Mergeay):

Associate Editor: 1

Comments to the Author:

Two reviewers have now evaluated your manuscript. Even though they agreed on the general quality of the research, they have additional questions and comments that require your attention.

I agree with reviewer 2 that the topic studied is complex, making some parts of the introduction and interpretation of the data hard to follow.

I think the paper would generally benefit from a graphical representation of the four scenarios, linking the processes acting upon the four scenarios with the resulting expected patterns of genetic variation and structure, and how to distinguish between them. It is described in the introduction (even though it wasn't entirely clear to reviewer 2), but a good schematic figure could help a lot. I actually found Fig S1 to be very illustrative, and suggest moving it to the main body of the paper: it represents the general phylogenetic relations among (taxa X sites), and suggests there is parallel ("independent") adaptation across different basins. Further digging into alternative hypotheses shows that there might be additional options, however.

Reviewer 1 remarks that the paper hardly deals with adaptations to an altitudinal gradient (even though it appears across an altitudinal gradient), or that this was not explicitly addressed. In addition, the genomic nature of the parallel adaptations (as reflected by the phenotypic data) was not clearly addressed. I agree that this doesn't seem to be the focus of the paper (contrary to what the title suggests), but rather the disentanglement of different processes (isolation, introgression, selection, ...) involved in the process of speciation across an ecological gradient. The exact nature of this gradient is less relevant. What is important is that it is spatially replicated across sites. The inclusion of the lake ecotypes distracted from this. Since these lake ecotypes represent three additional taxa, they seem to add an unnecessary complexity to the paper, without actually being well integrated into the core question (as they represent spatially unreplicated ecotypes).

The use of the word "genes" can be highly confusing too (e.g., line 41). Do you mean functional loci, or particular alleles on such loci under selection?

Also, mind the spelling, with regular typos occurring: Fig 4 mentions *S. richardsonii*, Fig 5 macrocephalus for example, but I also found here diversity instead of gene diversity.

Finally, please use the SI unit meter as the unit of distance (or elevation) instead of foot.

Best,
Joachim Mergeay

Reviewer comments to Author:

Reviewer: 1

Comments to the Author(s)

This study did a population genetics in polyploid snowtrout in *Schizothorax* with emphasis of species *S. progastus* and *S. richardsonii*. I found that the study in general is interesting and well conducted. I think that the main finding is that the authors found that *S. progastus* and *S. richardsonii* from Grandaki and Koshi genetically close to each other, which results from gene flow of secondary contact. However, I have the following concerns before its publication. First, I do not feel that analyses and results in the study really touch its topic of parallel adaptations to an altitudinal gradient in polyploid *Schizothorax* species. The results in Figure 4 and 5 do not completely reflect parallel genetic variation and the selection analysis does not touch either parallel genetic variation or parallel adaptations to an altitudinal gradient. Those results and analyses are only valid to gene flow of secondary contact between *S. progastus* and *S. richardsonii*. Second, since the study is working on polyploid species, it is important to consider read coverage when calling SNP. However, read coverage is not justified. It seems the authors used "6" that is too low for SNP calling in polyploid species. In addition, I have no idea what "transcriptome-guided assemblies" is. Third, there are several studies tried to work on population genetics of polyploid species, as the cited Meirmans et al. (2018). I am wondering how the results would be if analyses restrict to those tetraploid SNPs in Figure 2.

Reviewer: 2

Comments to the Author(s)

Review of Chafin_etal_ParallelAdaptationHimalayanFishes_RSOS

This paper employs a reduced representation library (modified ddRAD) approach to collect genome wide distributed SNP data for populations of Himalayan fishes to investigate parallel evolution. The paper then attempts to disentangle alternative hypotheses about the parallel evolution of adaptive ecotypes from either independent mutational events or a single shared mutation that subsequently spreads through gene flow followed by adaptive increases in allele frequency.

This is a complex system, with multiple nominal morphological species defined, as well as ancestral polyploidy complicating genetic inference. While generally well written, I found some sections to be difficult to understand. For example, the description of the alternative hypotheses and related expectations for the resulting genetic data described in the Introduction to be unclear. This seems further complicated by the models of how adaptive variation may spread through within-species metapopulation dynamics versus a model of speciation through adaptation. The former is more interactive, whereas the latter sort of presumes a directional movement towards speciation. I urge the authors to work on the clarity of the writing and to use consistent language throughout the text.

General Issues:

Poidy; The ancestral polyploidy of these species complicates the analysis, and the paper describes a 'ploidy-aware' genotyping pipeline. The description appears clear; however, I have not used this pipeline and did not validate their approach myself. More generally, polyploidy and rediploidization is a complex issue but the abstract states that they "unambiguously quantified ploidy levels...". Given the complexities I think this should be tempered somewhat.

>>Language, e.g. use of 'parallel' line 58, and 'independent' on Line 478. This is a crux point for understanding your interpretation, since parallel evolution from standing genetic variation could still be considered "independent" evolution of the phenotype but based on the same genetic variant rather than an independent new mutation. Please check your use of this language throughout the manuscript.

Finally I have found it impossible to review this manuscript without exposing my bias towards salmonid research and suggesting citations to my own papers, so will sign the review. I would be happy to clarify or discuss any aspects of this review, or to comment on a revised version of the manuscript. – Devon Pearse.

Line comments:

Abstract line 12; language "When genotyped SNPs were clustering...". The next sentence is also strangely worded-- if I understand correctly it is essentially describing an outlier analysis, but I don't think I understand the final part about genomic homogenization.

Intro:

Line 35: Here it is unclear what the contrasting hypotheses are-- What is the expectation of the relative importance of factors that might cause "genomic islands"?

>I can't follow the logic being described in the three paragraphs from Line 38-59. How do the alternative 'hypotheses' described lead to different predictions about patterns in the genetic data?

Also reference 29 Edelaar et al. 2008 seems like an old and not very relevant reference to cite and I am surprised by lack of references to much newer highly relevant work on parallel adaptive evolution and maintenance of 'genomic islands' especially in Salmonids, e.g. Larson et al. 2017; Pearse et al. 2014; 2019) as well as other fish taxa such as sticklebacks (Eda1; Jones et al. 2012) and Cod (genomic islands) as well as birds (e.g. Ruegg et al. 2014 Mol Ecol) and related discussion in the literature. Would be good to connect this paper to broader literature. Finally Line 59 ends by referring to the 'speciation process', but the processes being described really seem intraspecific in nature, occurring in metapopulations distributed in patchy habitat with alternative adaptive ecotypes.

Line 116; Sorry for even more self-promotion, but for comparison Campbell et al. 2019, G3, examined diploid and tetraploid ohnologs from a much older whole genome duplication in salmonids.

Line 126; Here I am not sure what "...deletion of historic ancestry..." means. Please clarify your language.

Methods

Line 300, Section (h): Dividing loci >1 or <1 for the ratio of Dxy divergence – seems like loci close to 1 really have no signal on which to base placing them in one group or the other. Should there be a gap in the thresholds above and below which they are partitioned?

Results

Figure 4b shows Bhutan genetic results, the widespread low elevation form is uniform. This result suggests that there is ongoing higher geneflow among populations of *S. prognathous* in the different watersheds, but less movement among *S. richarsonii* populations. This is very comparable to what happens in isolated trout populations, e.g. Pearse et al. 2014

Lines 370, 372; what are these numbers and substantial numbers? This is hard to infer from the figure, except for the 31 overlapping loci.

Discussion

Line 432; This is a clear statement of the authors' interpretation of the results and I agree with the way it is stated here except for the phrase 'subsequent homogenization...'. How was this shown to be true as opposed to alternative situations such as ongoing gene flow maintaining homogenization genome-wide except at adaptively important loci? Also Figure 7 doesn't appear to exist.

Line 435; 'rampant introgression', even a relatively small amount of ongoing gene flow will prevent genome wide divergence and maintain F_{st} near zero despite the presence of ecotypes maintained by specific adaptive genetic loci. See recent paper on run-timing ecotypes in salmon (Thompson et al. 2020 Science).

Lines 478-525; This section seems long and speculative and mostly repeats earlier conclusions. I suggest editing it down to just one or two paragraphs.

===PREPARING YOUR MANUSCRIPT===

===PREPARING YOUR REVISION IN SCHOLARONE===

- An individual file of each figure (EPS or print-quality PDF preferred [either format should be produced directly from original creation package], or original software format).
- An editable file of each table (.doc, .docx, .xls, .xlsx, or .csv).
- An editable file of all figure and table captions.

- Any electronic supplementary material (ESM).
- If you are requesting a discretionary waiver for the article processing charge, the waiver form must be included at this step.
- If you are providing image files for potential cover images, please upload these at this step, and inform the editorial office you have done so. You must hold the copyright to any image provided.
- A copy of your point-by-point response to referees and Editors. This will expedite the preparation of your proof.

- Ensure that your data access statement meets the requirements at <https://royalsociety.org/journals/authors/author-guidelines/#data>. You should ensure that you cite the dataset in your reference list. If you have deposited data etc in the Dryad repository, please include both the 'For publication' link and 'For review' link at this stage.
- If you are requesting an article processing charge waiver, you must select the relevant waiver option (if requesting a discretionary waiver, the form should have been uploaded at Step 3 'File upload' above).
- If you have uploaded ESM files, please ensure you follow the guidance at <https://royalsociety.org/journals/authors/author-guidelines/#supplementary-material> to include a suitable title and informative caption. An example of appropriate titling and captioning may be found at [https://figshare.com/articles/Table_S2_from_Is_there_a_trade-off_between_peak_performance_and_performance_breadth_across_temperatures_for_aerobic_sc](https://figshare.com/articles/Table_S2_from_Is_there_a_trade-off_between_peak_performance_and_performance_breadth_across_temperatures_for_aerobic_scope_in_teleost_fishes_/3843624) ope_in_teleost_fishes_/3843624.

Author's Response to Decision Letter for (RSOS-210727.R0)

See Appendix A.

Decision letter (RSOS-210727.R1)

Dear Dr Chafin

On behalf of the Editors, we are pleased to inform you that your Manuscript RSOS-210727.R1 "Parallel introgression, not recurrent emergence, explains apparent elevational ecotypes of polyploid Himalayan snowtrout" has been accepted for publication in Royal Society Open

Science subject to minor revision in accordance with the referees' reports. Please find the referees' comments along with any feedback from the Editors below my signature.

Please submit your revised manuscript and required files (see below) no later than 7 days from today's (ie 29-Sep-2021) date. Note: the ScholarOne system will 'lock' if submission of the revision is attempted 7 or more days after the deadline. If you do not think you will be able to meet this deadline please contact the editorial office immediately.

on behalf of Dr Joachim Mergeay (Associate Editor) and Kevin Padian (Subject Editor)
openscience@royalsociety.org

Associate Editor Comments to Author (Dr Joachim Mergeay):

Associate Editor

Comments to the Author:

Dear Dr Tyler,

The manuscript was greatly improved. I have a few extra comments and suggestions that you'll find in the attached file. I used the docx file and accepted all changes to have something without the visual clutter, and went through the entire manuscript to check for errors, clarity and made some notes, which I'd like you to check.

Sincerely,
Joachim Mergeay, associate editor

===PREPARING YOUR MANUSCRIPT===

Your revised paper should include the changes requested by the referees and Editors of your manuscript. You should provide two versions of this manuscript and both versions must be provided in an editable format:
one version identifying all the changes that have been made (for instance, in coloured highlight, in bold text, or tracked changes);

===PREPARING YOUR REVISION IN SCHOLARONE===

- If you are requesting a discretionary waiver for the article processing charge, the waiver form must be included at this step.
- If you are providing image files for potential cover images, please upload these at this step, and inform the editorial office you have done so. You must hold the copyright to any image provided.
- A copy of your point-by-point response to referees and Editors. This will expedite the preparation of your proof.

- Ensure that your data access statement meets the requirements at <https://royalsociety.org/journals/authors/author-guidelines/#data>. You should ensure that you cite the dataset in your reference list. If you have deposited data etc in the Dryad repository, please only include the 'For publication' link at this stage. You should remove the 'For review' link.
- If you are requesting an article processing charge waiver, you must select the relevant waiver option (if requesting a discretionary waiver, the form should have been uploaded at Step 3 'File upload' above).
- If you have uploaded ESM files, please ensure you follow the guidance at <https://royalsociety.org/journals/authors/author-guidelines/#supplementary-material> to include a suitable title and informative caption. An example of appropriate titling and captioning may be found at https://figshare.com/articles/Table_S2_from_Is_there_a_trade-off_between_peak_performance_and_performance_breadth_across_temperatures_for_aerobic_scope_in_teleost_fishes_/3843624.

Author's Response to Decision Letter for (RSOS-210727.R1)

See Appendix B.

Decision letter (RSOS-210727.R2)

Dear Dr Chafin,

I am pleased to inform you that your manuscript entitled "Parallel introgression, not recurrent emergence, explains apparent elevational ecotypes of polyploid Himalayan snowtrout" is now accepted for publication in Royal Society Open Science.

Please ensure that you send to the editorial office an editable version of your accepted manuscript, and individual files for each figure and table included in your manuscript. You can send these in a zip folder if more convenient. Failure to provide these files may delay the

processing of your proof. You may disregard this request if you have already provided these files to the editorial office.

on behalf of Dr Joachim Mergeay (Associate Editor) and Kevin Padian (Subject Editor)
openscience@royalsociety.org

Appendix A

>>>RESPONSE: We thank the Associate Editor and two reviewers for their constructive consideration of our submission. We have, to the best of our knowledge, addressed all of the direct suggestions either in-text, or where appropriate, as annotations to the original comments (below). This resulted in numerous minor changes, which we here summarize:

- 1) Terminology/ clarity: Multiple instances of inconsistent usage of terms (i.e. 'parallel', 'independent', 'gene' were pointed out, which we believe have been rectified in the revised MS. We have also modified the Introduction to now include a list of explicit objectives. We have also changed the title, removed superfluous material from the Discussion, and clarified details of our data handling (per Reviewer #1), in addition to minor modifications throughout.
- 2) Co-divergence: We have added a new analysis using the Bayesian program 'EcoEvolity', which allowed us to explicitly test the co-divergence of Lake Rara species, as well as other divergences in Nepal and Bhutan. This was previously speculated upon, but not actually tested.
- 3) New Figures in main document: Figure S1 has been moved to the main document, following the AE's suggestion, as it more clearly shows the general phylogenetic relationships than the combined Structure/ TreeMix figure. Also per the AE's suggestion, we have generated a figure illustrating the four scenarios described in the Introduction (now Figure 1).

We believe that this modified MS is much improved as a result – we hope the AE and reviewers are in agreement.

Editor comment:

Thanks for your submission. Although the reviewers recommend "accept with minor revision," in our experience their concerns merit "major revision" so that you can have the time to consider and respond to their issues. To this end we wish you the best, and please address their thoughts fully and individually. Thanks.

Associate Editor Comments to Author (Dr Joachim Mergeay):

Associate Editor: 1

Comments to the Author:

Two reviewers have now evaluated your manuscript. Even though they agreed on the general quality of the research, they have additional questions and comments that require your attention.

I agree with reviewer 2 that the topic studied is complex, making some parts of the introduction and interpretation of the data hard to follow.

I think the paper would generally benefit from a graphical representation of the four scenarios, linking the processes acting upon the four scenarios with the resulting expected patterns of genetic variation and structure, and how to distinguish between them. It is described in the introduction (even though it wasn't entirely clear to reviewer 2), but a good schematic figure could help a lot. I actually found Fig S1 to be very illustrative, and suggest moving it to the main body of the paper: it represents the general phylogenetic relations among (taxa X sites), and suggests there is parallel ("independent") adaptation across different basins. Further digging into alternative hypotheses shows that there might be additional options, however.

>>>RESPONSE: We have done as suggested and moved the former Figure S1 to the main body, which is now Figure 7. We've also attempted to further clarify the hypotheses/ objectives as they relate to these parallel groups by adding a list of explicit questions at the end of the Introduction.

We also attempted to create a schematic, with the goal of illustrating the four scenarios which could all (theoretically) result in the pattern of spatial, phylogenetic, and phenotypic discordance which could appear as ecotypy. Hopefully this has accomplished what the AE had in mind – if not, we could either remove it or modify it in a subsequent revision.

Reviewer 1 remarks that the paper hardly deals with adaptations to an altitudinal gradient (even though it appears across an altitudinal gradient), or that this was not explicitly addressed. In addition, the genomic nature of the parallel adaptations (as reflected by the phenotypic data) was not clearly addressed. I agree that this doesn't seem to be the focus of the paper (contrary to what the title suggests), but rather the disentanglement of different processes (isolation, introgression, selection, ...) involved in the process of speciation across an ecological gradient. The exact nature of this gradient is less relevant. What is important is that it is spatially replicated across sites. The inclusion of the lake ecotypes distracted from this. Since these lake ecotypes represent three additional taxa, they seem to add an unnecessary complexity to the paper, without actually being well integrated into the core question (as they represent spatially unreplicated ecotypes).

>>>RESPONSE: We've changed the title to more accurately depict the focus of the paper (i.e. on discriminating different types of parallel processes)

Lake types: We agree that we did a relatively poor job of describing our original intent with the Lake Rara samples. To better integrate these samples, we have 1) Added a list of objectives at the end of the Introduction; 2) Added a new analysis explicitly testing their co-divergence (alongside that of the riverine snowtrout pairs); and 3) Slightly modified the section in discussion dedicated to them (previously we speculated about recent/ rapid radiation, whereas now it has been tested).

The use of the word "genes" can be highly confusing too (e.g., line 41). Do you mean functional loci, or particular alleles on such loci under selection?

>>>RESPONSE: Revised use throughout in order to better clarify the intended meaning (excepting uses of "gene" as a part of accepted terminology e.g. "gene ontology" or "gene flow")

Also, mind the spelling, with regular typos occurring: Fig 4 mentions *S. richardsonii*, Fig 5 macrocephalus for example, but I also found hene diversity instead of gene diversity.

>>>RESPONSE: Corrected throughout

Finally, please use the SI unit meter as the unit of distance (or elevation) instead of foot.

>>>RESPONSE: Fixed.

Best,

Joachim Mergeay

Reviewer comments to Author:

Reviewer: 1

Comments to the Author(s)

This study did a population genetics in polyploid snowtrout in *Schizothorax* with emphasis of species *S. progastus* and *S. richardsonii*. I found that the study in general is interesting and well conducted. I think that the main finding is that the authors found that *S. progastus* and *S. richardsonii* from Grandaki and

Koshi genetically close to each other, which results from gene flow of secondary contact. However, I have the following concerns before its publication. First, I do not feel that analyses and results in the study really touch its topic of parallel adaptations to an altitudinal gradient in polyploid Schizothorax species. The results in Figure 4 and 5 do not completely reflect parallel genetic variation and the selection analysis does not touch either parallel genetic variation or parallel adaptations to an altitudinal gradient. Those results and analyses are only valid to gene flow of secondary contact between *S. progastus* and *S. richardsonii*.

>>>RESPONSE: We've changed the title to remove emphasis from the adaptations/ elevational gradient – as this reviewer correctly pointed out (as did the AE), these are not really central aspects of our paper.

Second, since the study is working on polyploid species, it is important to consider read coverage when calling SNP. However, read coverage is not justified. It seems the authors used "6" that is too low for SNP calling in polyploid species.

>>>RESPONSE: (Note that we separated some of the reviewer's comments here to better organize our response) This was only for the "pre-filtered" assembly which we did using ipyrad. Because the genotyping model in ipyrad is restricted to diploids, we only used it to assemble our loci (using the 'de novo' and 'transcriptome-guided' approaches), followed by genotyping in polyRAD, and then subsequent filtering requiring a minimum of 20X mean depth per locus to retain it. So, the final read coverage threshold was 20, not 6. We realize that the existing text failed to make this clear, so we've revised/moved some sentences around to better describe what was done.

In addition, I have no idea what "transcriptome-guided assemblies" is.

>>>RESPONSE: This assembly was done by mapping reads against the assembled transcriptome, as a contrast to the *de novo* assembly, which clustered reads irrespective of any existing reference information. Added a note in-text at first use of "transcriptome-guided" to clarify

Third, there are several studies tried to work on population genetics of polyploid species, as the cited Meirmans et al. (2018). I am wondering how the results would be if analyses restrict to those tetraploid SNPs in Figure 2.

>>>RESPONSE: Figure 2 shows the 'fit' of ploidy models across individuals, not across SNPs from nQuire, although we did also examine this across loci in polyRAD. Many analyses have restrictions as to the allowed 'ploidy', e.g. Adegnet DAPC. We did perform exploratory analyses of 'diploid-like' and 'tetraploid-like' loci in STRUCTURE and found no differences, so we didn't see it as necessary to explore this further with any of the analyses which allow flexible ploidy.

Reviewer: 2

Comments to the Author(s)

Review of Chafin_etal_ParallelAdaptationHimalayanFishes_RSOS

This paper employs a reduced representation library (modified ddRAD) approach to collect genome wide distributed SNP data for populations of Himalayan fishes to investigate parallel evolution. The paper then attempts to disentangle alternative hypotheses about the parallel evolution of adaptive ecotypes from either independent mutational events or a single shared mutation that subsequently spreads through gene flow followed by adaptive increases in allele frequency.

This is a complex system, with multiple nominal morphological species defined, as well as ancestral

polyploidy complicating genetic inference. While generally well written, I found some sections to be difficult to understand. For example, the description of the alternative hypotheses and related expectations for the resulting genetic data described in the Introduction to be unclear. This seems further complicated by the models of how adaptive variation may spread through within-species metapopulation dynamics versus a model of speciation through adaptation. The former is more interactive, whereas the latter sort of presumes a directional movement towards speciation. I urge the authors to work on the clarity of the writing and to use consistent language throughout the text.

General Issues:

Poidy; The ancestral polyploidy of these species complicates the analysis, and the paper describes a 'ploidy-aware' genotyping pipeline. The description appears clear; however, I have not used this pipeline and did not validate their approach myself. More generally, polyploidy and rediploidization is a complex issue but the abstract states that they "unambiguously quantified ploidy levels...". Given the complexities I think this should be tempered somewhat.

>>>RESPONSE: We've done as suggested and tempered this statement.

>>Language, e.g. use of 'parallel' line 58, and 'independent' on Line 478. This is a crux point for understanding your interpretation, since parallel evolution from standing genetic variation could still be considered "independent" evolution of the phenotype but based on the same genetic variant rather than an independent new mutation. Please check your use of this language throughout the manuscript.

>>>RESPONSE: We've tried to clarify as much as possible, as well as to remove uses of those terms in favor of synonyms where they do not explicitly refer to the biological scenarios in question, in an attempt to reduce confusion.

Finally I have found it impossible to review this manuscript without exposing my bias towards salmonid research and suggesting citations to my own papers, so will sign the review. I would be happy to clarify or discuss any aspects of this review, or to comment on a revised version of the manuscript. —Devon Pearce.

Line comments:

Abstract line 12; language "When genotyped SNPs were clustering...". The next sentence is also strangely worded-- if I understand correctly it is essentially describing an outlier analysis, but I don't think I understand the final part about genomic homogenization.

>>>RESPONSE: We agree that this sentence lost its meaning somewhat and have revised it. We've also made some other changes to the Abstract which we think improved the clarity.

Intro:

Line 35: Here it is unclear what the contrasting hypotheses are-- What is the expectation of the relative importance of factors that might cause "genomic islands"?

>>>RESPONSE: We don't offer any interpretation of the relative importance of alternative sources of genomic islands – just providing background that 'apparent' islands of divergence may originate from divergent selection against background gene flow (i.e. primary divergence) -or- purifying selection during secondary gene flow.

>I can't follow the logic being described in the three paragraphs from Line 38-59. How do the alternative 'hypotheses' described lead to different predictions about patterns in the genetic data? Also reference 29

Edelaar et al. 2008 seems like an old and not very relevant reference to cite and I am surprised by lack of references to much newer highly relevant work on parallel adaptive evolution and maintenance of 'genomic islands' especially in Salmonids, e.g. Larson et al. 2017; Pearse et al. 2014; 2019) as well as other fish taxa such as sticklebacks (Eda1; Jones et al. 2012) and Cod (genomic islands) as well as birds (e.g. Ruegg et al. 2014 Mol Ecol) and related discussion in the literature. Would be good to connect this paper to broader literature. Finally Line 59 ends by referring to the 'speciation process', but the processes being described really seem intraspecific in nature, occurring in metapopulations distributed in patchy habitat with alternative adaptive ecotypes.

>>>RESPONSE: Firstly, we've attempted to broaden the literature context within which the referenced section sits, both using some of the suggested literature and some new additions. Secondly, we've tried to more explicitly clarify that the content of this section is strictly limited to discussing alternative 'hypotheses' which could form a phylogenetic pattern consistent with 'independent parallel emergence' ... The subsequent section ("(a) Can parallel emergence be discriminated from parallel introgression?") focuses on how these hypotheses might be disentangled.

Changed "speciation" on Line 59 to "diversification", to better communicate the intended meaning irrespective of positioning along the speciation continuum

Line 116; Sorry for even more self-promotion, but for comparison Campbell et al. 2019, G3, examined diploid and tetraploid ohnologs from a much older whole genome duplication in salmonids.

>>>RESPONSE: We did end up citing this paper, but in the discussion.

Line 126; Here I am not sure what "...deletion of historic ancestry..." means. Please clarify your language.

>>>RESPONSE: Replaced with clearer wording

Methods

Line 300, Section (h): Dividing loci >1 or <1 for the ratio of Dxy divergence—seems like loci close to 1 really have no signal on which to base placing them in one group or the other. Should there be a gap in the thresholds above and below which they are partitioned?

>>>RESPONSE: Loci close to 1 could be those which contain little information, whether >1 or <1 , so we wouldn't expect the result to change much if there was an arbitrary gap – especially because what we actually saw in our data was that many of the loci >1.0 (i.e. those most important for this particular inference, i.e. fig 6a) are $\gg 1$ (fig. s3d), the distance among high elevation populations (*richardsonii* – *richardsonii*) vs. within drainage pairs (*richardsonii* – *progastus*) is exceedingly high. Given that these loci are the ones which contain the most information, and thus are driving the inferred phylogenetic relationships (fig. 6a), I don't think there is any reason to suspect that in practice the result should differ, and if anything should be more pronounced.

Results

Figure 4b shows Bhutan genetic results, the widespread low elevation form is uniform. This result suggests that there is ongoing higher geneflow among populations of *S. prognathous* in the different watersheds, but less movement among *S. richardsonii* populations. This is very comparable to what happens in isolated trout populations, e.g. Pearse et al. 2014

Lines 370, 372; what are these numbers and substantial numbers? This is hard to infer from the figure,

except for the 31 overlapping loci.

>>>RESPONSE: Added the numbers in text

Discussion

Line 432; This is a clear statement of the authors' interpretation of the results and I agree with the way it is stated here except for the phrase 'subsequent homogenization...'. How was this shown to be true as opposed to alternative situations such as ongoing gene flow maintaining homogenization genome-wide except at adaptively important loci? Also Figure 7 doesn't appear to exist.

>>>RESPONSE: We removed the word "subsequent" in order to make the statement more generic to include any form of gene-flow mediated homogenization

Line 435; 'rampant introgression', even a relatively small amount of ongoing gene flow will prevent genome wide divergence and maintain F_{st} near zero despite the presence of ecotypes maintained by specific adaptive genetic loci. See recent paper on run-timing ecotypes in salmon (Thompson et al. 2020 Science).

>>>RESPONSE: Thanks for the suggestion. We cited this paper as an statement in the Intro on how discriminating among different 'scenarios' behind the origin of ecotypes is complicated by the fact that the genetic architecture of ecotype adaptations can be highly localized.

Lines 478-525; This section seems long and speculative and mostly repeats earlier conclusions. I suggest editing it down to just one or two paragraphs.

>>>RESPONSE: We removed entirely the first paragraph of this section, which related to details which are likely unimportant for the majority of readers. We also attempted to make the remainder more concise, and also removed the last several sentences. The new section (c) is considerably shorter.

Appendix B

Parallel introgression, not recurrent emergence, explains apparent elevational ecotypes of polyploid Himalayan snowtrout

T.K. Chafin, B. Regmi, M.R. Douglas, D.R. Edds, K. Wangchuk, S. Dorji, P. Norbu, S. Norbu, C. Changlu, G.P. Khanal, S. Tshering, and M.E. Douglas

Response to Reviewers

Dear Editorial Staff:

We thank the Associate Editor for their comments on the revised version of this manuscript. Several errors were pointed out, as well as suggestions to improve the clarity – we believe these to have been fully addressed in this final draft, which we present in hopes of acceptance in *Royal Society Open Science*.

Thank you again for a speedy and productive review process.

- Chafin et al.

--

Associate Editor Comments to Author (Dr Joachim Mergeay):

Associate Editor

Comments to the Author:

Dear Dr Tyler,

The manuscript was greatly improved. I have a few extra comments and suggestions that you'll find in the attached file. I used the docx file and accepted all changes to have something without the visual clutter, and went through the entire manuscript to check for errors, clarity and made some notes, which I'd like you to check.

Sincerely,

Joachim Mergeay, associate editor